# Do We Need TNM for Tracheal Cancers? Analysis of a Large Retrospective Series of Tracheal Tumors

**DOI:** 10.3390/cancers14071665

**Published:** 2022-03-25

**Authors:** Aleksandra Piórek, Adam Płużański, Paweł Teterycz, Dariusz Mirosław Kowalski, Maciej Krzakowski

**Affiliations:** 1Department of Lung Cancer and Thoracic Tumors, Maria Sklodowska-Curie National Research Institute of Oncology, 02-781 Warsaw, Poland; adam.pluzanski@pib-nio.pl (A.P.); dariusz.kowalski@pib-nio.pl (D.M.K.); maciej.krzakowski@pib-nio.pl (M.K.); 2Department of Soft Tissue/Bone Sarcoma and Melanoma, Maria Sklodowska-Curie National Research Institute of Oncology, 02-781 Warsaw, Poland; pawel.teterycz@pib-nio.pl; 3Department of Computational Oncology, Maria Sklodowska-Curie National Research Institute of Oncology, 02-781 Warsaw, Poland

**Keywords:** tracheal tumors, adenoid cystic carcinoma of the trachea, squamous cell carcinoma of the trachea, staging systems, classification, treatment

## Abstract

**Simple Summary:**

The TNM (tumor, node, metastases) staging system established by the Union for International Cancer Control/American Joint Committee on Cancer is commonly used to select a treatment method for patients with cancers, guide adjuvant therapy after surgery, and predict the prognosis. It is an essential tool for working with cancer patients in everyday medical practice. Currently, the eighth edition of the TNM staging system is used. Primary tracheal tumors are uncommon neoplasms. Probably due to their rarity, neither an AJCC staging system nor other, widely accepted staging system exists for primary tracheal cancers. Only a few studies stated their proposed guidelines for the staging of tracheal neoplasms. The absence of a universally adopted staging system makes it difficult for clinicians to assess tracheal cancers properly. This makes it challenging to conduct analyses and compare the results of published works. A standard classification system would help assess and qualify patients for treatment and, perhaps, establish uniform indications for adjuvant treatment. All this could contribute to an increase in the proportion of patients qualified for radical surgery, which is the preferred treatment method.

**Abstract:**

Due to the low incidence of primary tracheal neoplasms, there is no uniform system for staging of this disease. Our retrospective analysis based on registry data included 89 patients diagnosed with primary tracheal cancer at the National Research Institute of Oncology in Warsaw, Poland, between January 2000 and December 2016. We analyzed demographic, clinical, pathological, therapeutic, and survival data. The staging—for the purpose of our analysis—was performed retrospectively on the basis of imaging results. Tumor (T) category was defined as a disease confined to the trachea or lesion derived from the trachea and spreading to adjacent structures and organs. Node (N) and metastases (M) categories were divided into absence/presence of metastasis in regional lymph nodes and the absence/presence of distant metastasis. Survival analysis was performed depending on the clinical presentation of these features. There was a significant difference in overall survival depending on the T, N, M categories in the entire group. In the group of patients undergoing radical treatment, the T and N categories had a statistically significant impact on overall survival. In the group of patients treated with palliative aim, only the T category had an impact on overall survival. Multivariate analysis showed statistical significance for the T category in patients undergoing radical and those receiving palliative treatment. The assessment of the anatomical extent of lesions may help decide about treatment options and prognosis.

## 1. Introduction

Primary tracheal neoplasms constitute 0.2% of all cancers of the respiratory tract and 0.02–0.04% of all malignant neoplasms [1]. The annual incidence is about 0.1 cases per 100,000. About 90% of primary tracheal tumors in adults are malignant [1]. Squamous cell carcinoma (SCC) and adenoid cystic carcinoma (ACC) together account for more than two-thirds of primary tracheal cancers in adults [2].

The diagnosis is usually late due to the large functional reserve of the tracheal lumen. The first symptoms appear only when the tracheal lumen is reduced by 50–75%. Exertional dyspnea appears when the lumen is narrowed to 8 mm, and resting dyspnea appears when it is narrowed to 5 mm [2,3]. The presented symptoms are nonspecific and may lead to the misdiagnosis of asthma, chronic obstructive pulmonary disease, or bronchitis. The most common symptom of tracheal SCC is hemoptysis. The occurrence of hemoptysis usually leads to an earlier diagnosis of the neoplasm. However, hemoptysis occurs in <25% of patients in the early stages of the disease. The lack of symptoms often delays diagnosis by up to several months [2]. The presence of hoarseness and dyspnea usually indicates an advanced disease. Wheezing and stridor are the most common symptoms of adenoid cystic carcinoma.

Radical surgery is the treatment of first choice whenever the stage of the disease allows it. The scope and type of surgery depend on the location and size of the primary tumor and the involvement of adjacent structures [2,3,4,5].

Tracheal neoplasms—due to their rarity—are not included in the “tumor, node, metastases” (TNM) classification of malignant neoplasms. There are only (not prospectively confirmed) classification proposals describing the anatomical extent of the disease, which are unfortunately based on a small number of assessed patients [2,4,6,7,8,9]. The assessment of the anatomical extent of the lesions may be helpful in making a decision about the choice of treatment method and may be of prognostic value. Therefore, in this study, we aimed to examine the prognostic significance of TNM in patients with primary tracheal tumor.

## 2. Materials and Methods

The retrospective analysis included patients treated at the Maria Sklodowska-Curie National Research Institute of Oncology in Warsaw between January 2000 and December 2016. Patients with tracheal cancer were identified by searching the institution’s cancer registry. Adult patients diagnosed with primary cancer of the trachea (excluding patients with tumors that could originate in the larynx, main bronchus, or sites outside the trachea, such as the thyroid or esophagus) were included in the analysis (Figure 1).

Eighty-nine actively treated patients diagnosed with primary tracheal cancer were identified. Demographic, clinical, and pathological data (symptoms, smoking history, performance status, histological diagnosis, location, extent of the disease) and treatment information (purpose, modality) were obtained from traditional and electronic medical records. Because tracheal neoplasms are not included in the International Union Against Cancer (UICC) and American Joint Committee on Cancer (AJCC) classification systems, the staging was performed retrospectively on the basis of available imaging results before qualification for treatment. The location and the extent of the disease were estimated from the baseline computed tomography scans and descriptions of bronchoscopic examinations when available. Positron emission tomography–computed tomography examination was only available in individual cases. The T category was defined as a tumor confined to the trachea or a tumor originating in the trachea that spreads beyond the trachea to adjacent structures and organs. The N and M categories were divided dichotomously into the following: the absence or presence of metastasis in the regional lymph nodes and the absence or presence of a distant metastasis (Table 1). The stages of the disease were defined as follows:localized disease when the tumor was confined to the trachea;locoregional disease, when the tumor extended beyond the trachea to adjacent structures and/or the lymph nodes are involved;disseminated disease, when the cancer has spread to distant organs.

**Table 1 cancers-14-01665-t001:** Adopted clinical TNM classification.

T—Primary Tumor	
T1	Tumor confined to the trachea
T2	A tumor originating in the trachea that spreads beyond the trachea to adjacent structures and organs ^1^
N—Regional Lymph Nodes	
N0	No metastasis in the regional lymph nodes
N1	Regional lymph node metastasis present
M—Distant Metastases	
M0	Distant metastasis absent
M1	Distant metastasis present

^1^ Tumor growing through the tracheal wall +/− including larynx or carina +/− growing into the mediastinum and spread to neighboring organs or structures.

Due to the lack of a uniform staging system for tracheal neoplasms and significant heterogeneity in the group of patients with advanced locoregional disease, in the proposed classification, it was decided to perform statistical analyses for specific TNM categories instead of the proposed cancer stages.

The observation was completed on 31 December 2019. Information on survival was obtained from the medical records and from the offices keeping the records of the population movement. Patients were determined to have died if their name, date of birth, and PESEL (Universal Electronic System for Registration of the Population) number matched. The median follow-up was 93.4 months (95% CI: 76.4–NR). At the time of analysis, 15 patients (17%) were alive (including two who ended follow-up after >10 years of observation). The follow-up examinations included a computed tomography scan every 3 months for the first year and then every 6 months. Bronchoscopy was performed when necessary. The overall survival (OS) was calculated from the date of the first diagnosis until death from any cause. The study was performed according to the Helsinki Declaration and the Institutional Review Board Committee.

Patient demographics, tumor characteristics, and details of treatment and tumor response were summarized using the number of patients and percentages of the whole group. The differences between groups were assessed using the Mann–Whitney U test. The Kaplan–Meier method for estimating survival functions and the Cox proportional hazards model for estimating the effects of covariates on the hazard of the occurrence of death were used. All confidence intervals (CI) were 95%. All *p*-values <0.05 were considered significant. No adjustment for multiple testing was performed. All analyses were performed in the R language environment version 3.5.1 (The R Foundation for Statistical Computing, Vienna, Austria).

## 3. Results

The most frequent histological type was SCC, which was diagnosed in 50 out of 89 patients (56.2%), while ACC was found in 19 patients (21.3%). The remaining histological diagnoses were grouped for statistical purposes as “other” and were not subsequently differentiated. The histological distribution of all cancers, including a group of other histological types, is shown in Table 2. The distribution of demographic and clinical data by histological type is summarized in Table 3.

Seventy-eight percent of patients diagnosed with SCC were over the age of 60 years. The tumor did not occur in the age group under 35 years of age. ACC was diagnosed in various age groups (36.8% of patients under 35 years of age). SCC was diagnosed more often in men (66%), while ACC was more often diagnosed in women (73.7%). Of the 43 patients for whom smoking history was available, 100% of patients diagnosed with SCC were current or former smokers.

The most frequently reported symptoms were dyspnea (37.1%) and hemoptysis (36%). Hemoptysis was the first symptom of disease in 48% of patients diagnosed with SCC and 10.5% of patients with ACC. The distribution of the T feature varied among the histological types. Lymph node metastases were more frequent in SCC than in ACC. Distant metastases were detected in 13.5% of patients at diagnosis. The highest stages of the disease were observed in the group of other histological types.

In the presented group, 45 patients underwent primary radical treatment, and 44 were qualified for exclusive palliative treatment. The clinical staging in these two groups is presented in Table 4.

Surgical resection was performed in 13 patients (28.9%) out of 45 radically treated patients (10—ACC, 1—SCC, and 2—other histological type). Radiotherapy alone as the primary method of radical treatment was used in 25 patients (55.5%). The distribution among the histological types was as follows: 6—ACC, 15—SCC, 4—other. Radiotherapy was the most common type of treatment used in the group of patients treated with a palliative intention—33 patients (74.9%). The remaining patients were treated with chemotherapy (15.9%) or surgical palliative treatment aimed at restoring the airways (9.1%).

The median OS in the analyzed group with the 95% confidence interval (CI) was 13.3 months (range: 9.2–26.2 months). The proportion of 5 year OS in the entire analyzed group was 24.2% (95% CI, 16.7%–35.2%). The 5 year OS rates in the group of patients who underwent radical treatment and in the group of patients who underwent palliative treatment was 45.9% and 2.3%, respectively (*p* < 0.001). The median OS in these two groups was 46.1 months and 7.2 months, respectively. In the group of patients undergoing primary surgical treatment with radical intention, the 5 year OS was 76.9% compared to 35.8% in the group of patients undergoing nonsurgical treatment.

Survival analysis was performed depending on the clinical presentation of T, N, and M features. There was a significant difference in OS depending on the T category in the entire analyzed group of patients (46.1 vs. 8.8 months, *p* < 0.001). There was a statistically significant difference between OS among people with or without lymph node involvement (37.7 vs. 8.7 months, *p* < 0.001). The relationship between OS and the presence of metastatic lesions at the diagnosis of the disease was analyzed. It was shown that the presence of distant metastases was a negative prognostic factor for the entire analyzed group (17.7 vs. 6.8 months, *p* = 0.012). Cumulative probability of overall survival in the entire analyzed group of patients according to TNM categories is presented in Figure 2. Additional analysis of the cumulative probability of overall survival according to histological type by TNM category is shown in Figure 3.

In the group of patients undergoing radical treatment, the differences in the T and N categories had an impact on OS. Among patients treated with radical intention, the 5 year OS in patients with T1 feature was 74.5%, and, in patients with T2 feature, it was 24.3%. The 5 year OS in patients with N0 and N1 categories was 65% and 0%, respectively, and the 3 year OS was 73.1% and 16.7%, respectively. In the group of patients treated with palliative care, only differences with regard to the T category were observed. The 5 year OS in patients with T1 and T2 categories was 0% and 4.4%, respectively. On the other hand, the 3-year OS was 22.2% and 4.4%, respectively, and the 1 year OS was 55.6% and 8.7%, respectively. Figure 4 and Figure 5 show the overall survival curves according to T and N depending on the treatment intention. Multivariate analysis revealed statistical significance for T feature both in the group of patients undergoing radical and in those receiving palliative treatment (Figure 6 and Figure 7).

According to multivariate analysis, we identified also the following prognostic factors for overall survival: gender, histological type of tumor and performance status in radically treated patients, and histological type of tumor for patients treated palliatively. In the group of patients receiving palliative treatment, gender and performance status had no effect on prognosis.

## 4. Discussion

Due to the low incidence of primary tracheal neoplasms, there is no uniform staging system. The publications proposing a staging method in primary tracheal carcinoma are presented below.

The first staging system, suggested by Licht et al., was presented in 2001 [6]. The authors assessed the location and size of lesions, as well as the presence of lesions in distant organs, on the basis of conventional X-ray examinations, as well as computed tomography (CT), magnetic resonance imaging (MRI), and endoscopic examination reports. In patients treated in the years 1978–1995, the following criteria were used: (1) TI: tumor <3 cm, not exceeding the tracheal wall, without larynx and carina involvement; TII: tumor <3 cm growing beyond the tracheal wall, but without involvement of nearby organs and without involvement of the larynx and carina; TIII: tumor >3 cm, involving the larynx and/or growing into mediastinum, but without invasion of adjacent organs; TIV: tumor >3 cm involving the carina and/or main bronchus or involving adjacent organs; (2) N0: no lymph node metastases; NI: metastases in regional lymph nodes; NII: metastases in distant lymph nodes; (3) M0: distant metastases absent; MI: distant metastases present. The stage of the disease was presented on the basis of the following criteria: stage I: TI + N0 + M0; stage II: TII–III + N0 + M0; stage III: TI–III + NI + M0; stage IV: TIV or NII or MI. The stage distribution among 92 patients was as follows: stage I in 21%, stage II in 23%, stage III in 6%, and stage IV in 50%. Distant metastases were present in eight cases (9%). The tumor infiltrated the esophagus in six patients (7%) and the thyroid gland in three patients (3%), while large blood vessels were involved in seven cases (8%). In 17 patients (18%) the tumor involved the main bronchus, but it was considered the primary tumor of the trachea. It was shown that patients with stage I have better survival than patients with stages II–IV.

Bhattacharyya presented a classification system based on the evaluation of 92 patients with tracheal cancer over a period of 12 years (1988–2000) [7]. It is the most frequently quoted system for the staging of tracheal neoplasms. In the study, the T category divided the size (limited to the trachea, <2 or >2 cm—T1, T2) and the extent of the primary tumor (the lesion spreading beyond the trachea, infiltrating or not adjacent organs and structures—T4, T3). The N category simply defined the presence or absence of metastasis in the regional lymph nodes. The stages were grouped as follows: I: T1N0; II: T2N0; III: T3N0; and IV: T4N0 or T1–T4N1. Affected lymph nodes were found in 20.7% of patients. Fifty-three percent of patients were in stage III and IV. The percentages of 5 year survival were as follows: I—52.5%, II—70.0%, III—75.0%, and IV—15.1%. Worse results for stage I were explained by the greater number of patients diagnosed with SCC in this group.

A more comprehensive system, referring to the TNM classification system for head and neck cancers, was proposed by Paolo Macchiarini in 2006 [2]. The prognostic significance of the disease stage was not assessed in this study. In the same year, a system based on the Bhattacharyya classification but taking into account the M feature was published by Webb et al. The outcomes were better in patients with small tumors, without lymph node metastases and without distant metastases [4].

The Polish system was proposed by a team from the Krakow branch of the Institute of Oncology. Stage I included disease confined to the trachea, stage II included those confined to the chest, stages IIIA and IIIB had additional involvement of regional and supraclavicular lymph nodes, and stage IV included the presence of distant metastases. The 5 year survival rates for patients with subsequent stages of the disease were 60%, 14%, 12%, 0%, and 0%, respectively. The authors made an interesting comparison with the Bhattacharyya classification. The stage of the disease was an independent prognostic factor for OS [8].

In the latest study, published in 2018, in which the classification system was based on data obtained from the SEER database, the authors categorized patients taking into account the extent of disease based on the involvement of adjacent structures (including individual vessels, nerves, and adjacent organs), primary tumor size, lymph node involvement, and the presence of distant metastases [9]. Lymph node metastases and distant metastases were independent factors affecting patient survival.

In our study a modified disease stage classification system was used. The T category was defined as a tumor confined to the trachea or a tumor originating in the trachea that spreads beyond the trachea to adjacent structures and organs. N and M categories were divided into the absence or presence of metastasis in the regional lymph nodes and distant metastases. In the TNM classification of malignant neoplasms, the staging frames are adapted to each tumor location in order to ensure the homogeneity of each group in terms of survival rates and differentiation of survival between groups of different stages [10]. Due to the lack of a uniform stage grouping system in tracheal neoplasms and significant heterogeneity in the group of patients with locally advanced disease, in the proposed classification, it was decided to carry out analyses in relation to specific TNM categories instead of the proposed cancer stages. A literature review was carried out, focusing on the assessment of the significance of the analyzed T, N, and M categories, and it was compared with the authors’ own results.

### 4.1. T Category

In the literature, there are various means of assessment of primary tracheal cancer. Some studies take into account its size, some take into account its extent, and others take into account both elements. Due to the retrospective nature of our study, the size of the tumor was difficult to unequivocally assess. We focused on the extent with the assessment of the involvement of adjacent structures and organs. It was shown that the presence of a tumor confined to the trachea positively affects all the assessed survival parameters. Similar results were obtained in the study of Wen et al. [9]. It has been shown that the extent of the primary tumor is an independent factor affecting OS in all groups of patients, which was not confirmed in the above-cited study. In other studies assessing the extent of the disease, it was shown that patients with involvement of adjacent structures and organs have an unfavorable prognosis similarly to patients with affected lymph nodes [7]. It has also been shown that thyroid infiltration in the case of SCC is a negative prognostic factor [11] and that the T feature is a significant prognostic factor in airway ACC [12]. Additionally, some studies showed the prognostic significance of tumor size [9,13], but others did not confirm this observation [12,14,15].

### 4.2. N Category

In the presented study, in a group of 71 evaluable patients, lymphadenopathy was found in 42.3% (ACC—2.8%, SCC—29.6%). In the total group of patients and in the group of patients treated radically, a significantly worse 5 year OS was demonstrated in the case of clinical lymph node involvement (general group 41.2% vs. 0%; radically treated group 65% vs. 0%). These results are consistent with the literature reports [4,7,8,9]. In the study by Bhattacharyya et al., the authors showed that the survival of patients with positive regional lymph nodes was more than 50% worse compared to patients with the N0 feature [7].

Many studies emphasized that the risk of lymph node involvement, and the difference in survival associated with it differs depending on the histological type of the tumor. For a detailed comparison, studies were found in which the pathomorphological verification of lymph nodes was also carried out in the case of various histological types of cancer. In the case of ACC, lymph node involvement was found in 0–35.3% of cases and was defined as a negative prognostic factor in most studies. The location of ACC in the bronchus versus the trachea (*p* = 0.001) and size of tumor greater than 3 cm (*p* = 0.003) were identified as independent risk factors for the occurrence of lymph node metastases [16]. In a large study involving patients diagnosed with ACC treated in the years 1962–2007, the 5 year OS rates were 76% in patients with N0 and 54% with pathologically confirmed involved lymph nodes (*p* = 0.017) [17]. However, in a study from the same center on patients diagnosed with SCC, the 5 year OS was 60% and 24%, respectively (*p* = 0.049) [11]. In a study that evaluated the status of lymph nodes after 191 resections, lymph node metastasis was found in 19.4% of patients (nodal biopsies were not obtained in 35%), most commonly from peritracheal and subcarinal stations. One positive lymph node was diagnosed in 24 patients (16 SCC patients and eight ACC patients). More than one positive lymph node was found in eight and five patients, respectively. Univariate analysis revealed a decrease in survival for patients with positive lymph nodes in SCC. However, the authors did not confirm this in multivariate analysis [14]. Studies in which the effect of lymph node involvement on survival was not shown drew attention to the high percentage of patients with lymph node involvement, receiving adjuvant radiotherapy after surgery [14,18]. Due to limitations resulting from the small group of surgically treated patients at our institute, it was not possible to assess the impact of pathologically confirmed involved lymph nodes in the current study.

### 4.3. M Category

Distant metastases at diagnosis of tracheal cancer are rare. Gaissert et al. observed distant metastases in about 10% of patients with ACC and SCC of the trachea at the time of diagnosis [9]. Few studies have assessed the prognostic significance of the M category. The two largest studies, in which a representative number of patients with metastases were taken into account, showed a statistically significant negative effect on OS [9,16]. In the presented study, synchronous distant metastases were found in 12 (13.5%) patients (ACC—2, SCC—4, other type—6). The lungs were the most common location of metastases. The above observations regarding the percentage of patients with distant metastases in particular histological types of cancer and the most frequently involved organs are similar to the data from the available literature. In the present study, the M category was also statistically significantly associated with OS.

It should be added that ACC has distinct growth dynamics and time to appearance of metastatic lesions. In a study from 1996, Maziak et al. found only metachronic distant metastases in more than half of the assessed patients, and the lungs were the most frequent location. The authors showed that these lesions often appear long after initial diagnosis (12–300 months) and that patients with late dissemination have long survival up to 7 years after diagnosis (mean survival time 37 months). They confirmed the lack of evidence of the effectiveness of chemotherapy in this group of patients [19]. In various studies, the authors also emphasized that lung metastases in ACC may remain asymptomatic for a long time, emphasizing the role of surgical treatment and the need for a long follow-up [12,15,20].

The TNM classification system of malignant tumors does not include tracheal neoplasms due to their low incidence. Therefore, data analysis is made difficult by the differences between the presented classifications. More importantly, however, the decision-making process is hampered when selecting patients for a specific treatment, as is the assessment of prognosis. A disease staging system has been shown to have prognostic value. Although the presented article evaluated a larger series of patients with primary tracheal neoplasms, the study had its limitations. Firstly, it was based on the analysis of a small and heterogeneous group of patients treated for 7 years. Secondly, most of the patients were diagnosed outside our center. Thirty-five percent of the patients started treatment in another hospital and were only referred to our institute for adjuvant therapy, treatment of recurrence, or observation; thus, the data were incomplete. Nevertheless, the treatment of all patients was verified at multidisciplinary meetings. We could not assess baseline performance status, symptoms, or smoking status in all patients. The tumor staging alone was based mainly on the review of computed tomography scans and available endoscopy reports. For individual patients, positron emission tomography and magnetic resonance imaging results were available. We were unable to consider and evaluate the effect of tumor size. Furthermore, the classification does not include the pathological verification of TNM due to the small number of surgically treated patients. Undoubtedly, an advantage of our study is that it evaluated groups of patients treated with radical and palliative intent. In most literature studies, these patient groups were evaluated together. However, due to the small groups of patients, we were unable to perform TNM analysis according to the number of lymph nodes involved or the number of distant metastases, and the analysis by histological type should be treated with caution. It is well known that SCC of the trachea and ACC have different clinical course; however, tracheal tumors include other histological types (22.5% in the presented study) that are little known and rarely described in research papers. High-quality data for such rare diseases are hard to obtain, and a prospective study is very difficult to conduct. Nevertheless, TNM assessment by histological type and analysis of tracheal neoplasms other than SCC and ACC represent interesting research directions.

## 5. Conclusions

Despite its limitations, the proposed simple classification according to TNM allowed us to distinguish groups of patients with favorable prognosis. The proposed classification system allows for the identification of patient groups by treatment intent. For this group of patients, we emphasize the need for a centralized care system in centers with surgery and radiotherapy facilities, with access to systemic treatment and experience in treating this rare disease. This would give a chance of early diagnosis and a possibility of radical treatment. We stress how important it is to verify the extent of the tumor and assess resectability, preferably at multidisciplinary meetings. Answering the study’s title question, we note the need to create a multicenter database for primary tracheal tumors, which should be developed in cooperation with national and international cancer centers and scientific societies. This would allow homogeneous data to be obtained and would facilitate the development of uniform treatment protocols based on a well-defined spread of disease.

## Figures and Tables

**Figure 1 cancers-14-01665-f001:**
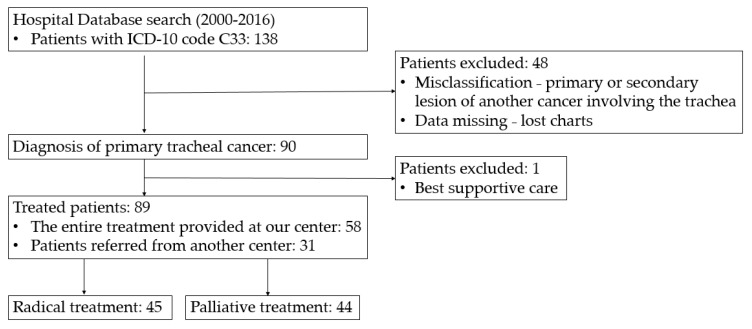
Patient identification algorithm and exclusion criteria.

**Figure 2 cancers-14-01665-f002:**
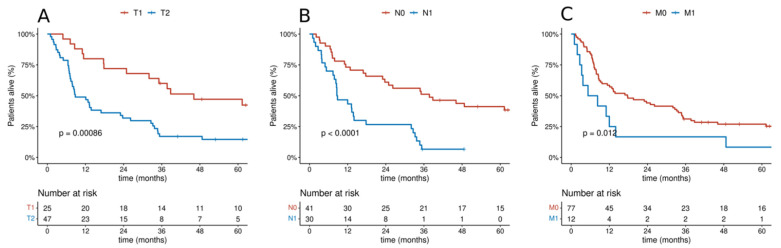
Cumulative probability of overall survival in the entire analyzed group of patients according to categories: (**A**) T; (**B**) N; (**C**) M.

**Figure 3 cancers-14-01665-f003:**
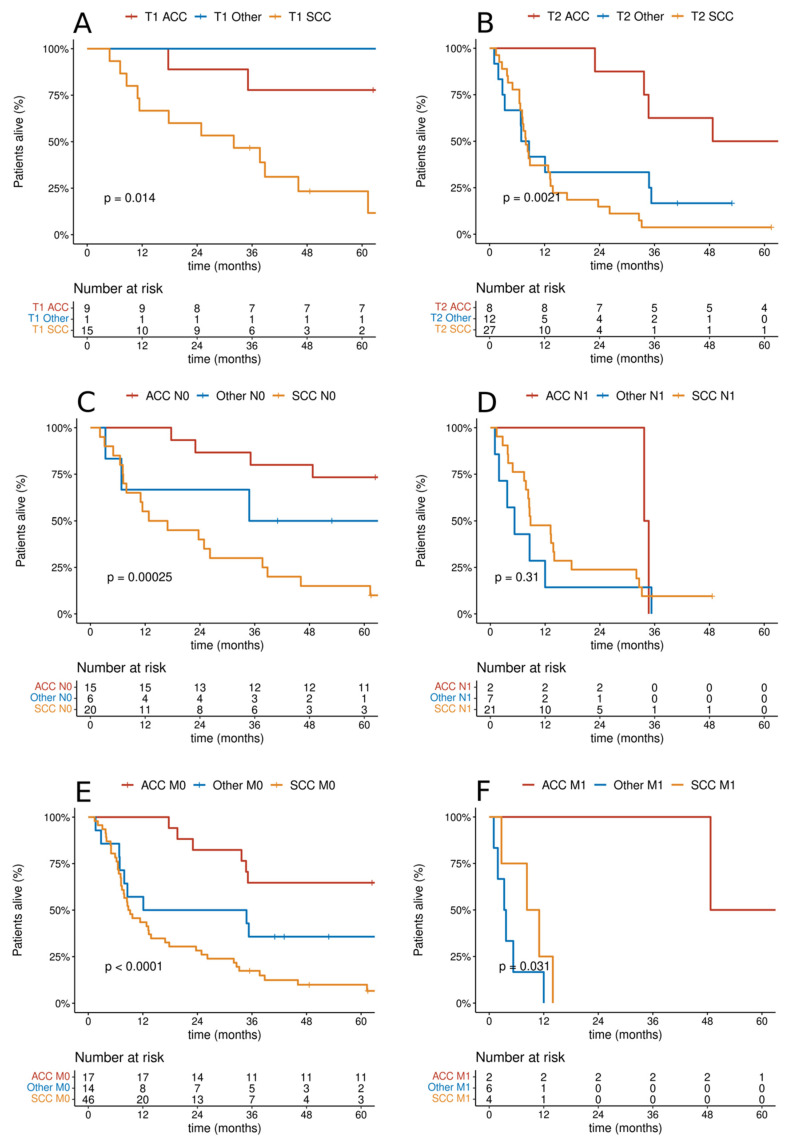
Cumulative probability of overall survival according to histological type by TNM category: (**A**,**B**) T1/T2; (**C**,**D**) N0/N1; (**E**,**F**) M0/M1.

**Figure 4 cancers-14-01665-f004:**
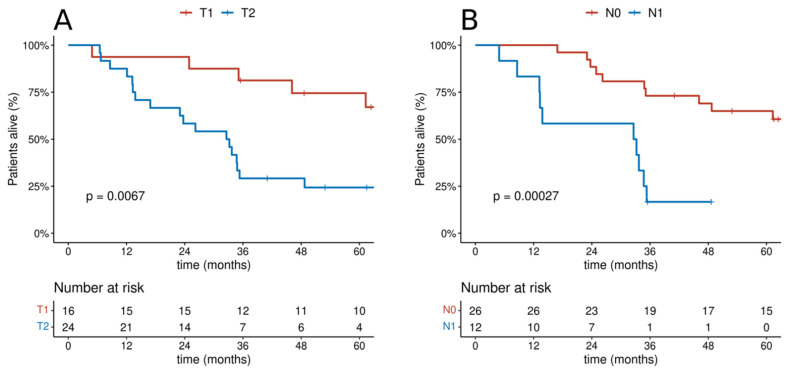
Cumulative probability of overall survival in the group of patients receiving radical treatment according to categories: (**A**) T; (**B**) N.

**Figure 5 cancers-14-01665-f005:**
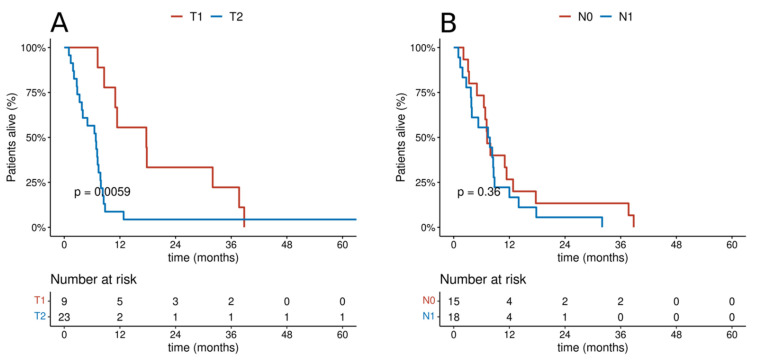
Cumulative probability of overall survival in the group of patients receiving palliative treatment according to categories: (**A**) T; (**B**) N.

**Figure 6 cancers-14-01665-f006:**
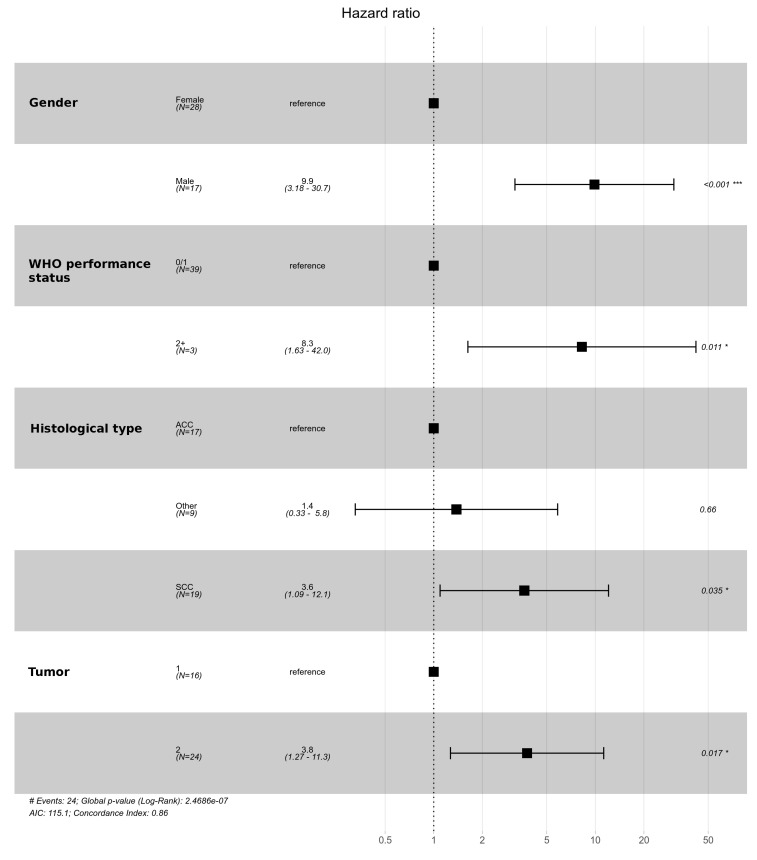
Multivariate analysis for overall survival among radically treated patients. Significant codes legend: from 0 to 0.001 “***”; from 0.01 to 0.05 “*”.

**Figure 7 cancers-14-01665-f007:**
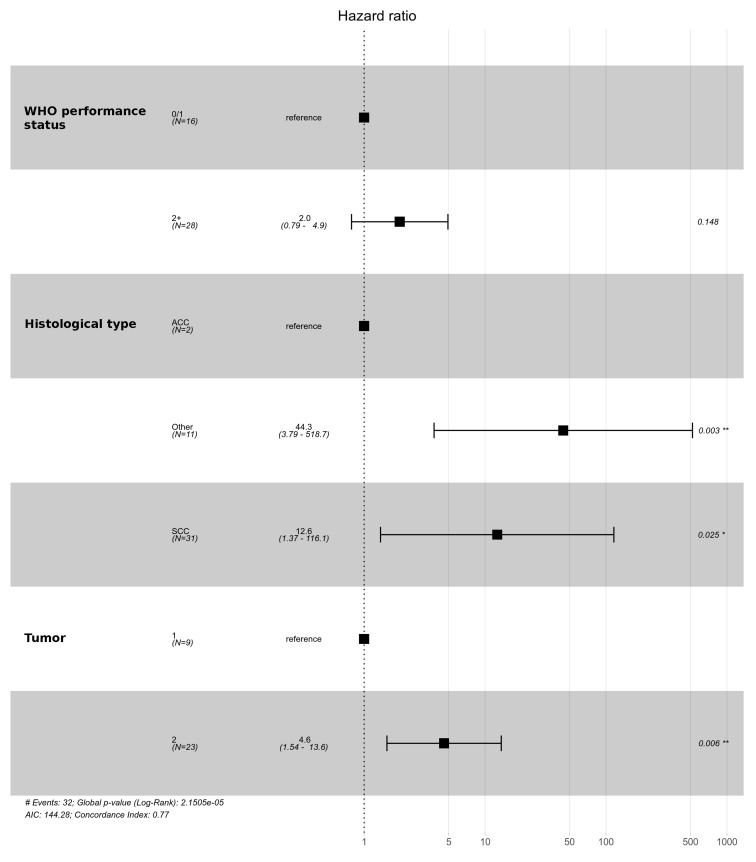
Multivariate analysis for overall survival among palliative care patients. Significant codes legend: from 0.001 to 0.01 “**”; from 0.01 to 0.05 “*”.

**Table 2 cancers-14-01665-t002:** Histological distribution of 89 tracheal cancers.

Histology	*n*	%
Squamous cell carcinoma	50	56.2
Adenoid cystic carcinoma	19	21.3
Other	20	22.5
Non-small-cell carcinoma	12	13.5
Adenocarcinoma	4	4.5
Malignant peripheral nerve sheath tumor	2	2.2
Small-cell carcinoma	1	1.1
Unspecified	1	1.1

**Table 3 cancers-14-01665-t003:** Demographic and clinical characteristics of 89 patients diagnosed with primary tracheal neoplasm by histological type.

Clinicopathological Factor			ACC	Other	SCC	*p* ^1^
Number of patients			19	20	50	
Age—median (range)			43.00 (28.50–56.00)	62.50 (51.00–74.25)	63.50 (58.25–68.00)	<0.001
Gender (%)		Female	14 (73.7)	10 (50.0)	17 (34.0)	0.012
		Male	5 (26.3)	10 (50.0)	33 (66.0)	
WHO performance status (%)		0	6 (35.3)	5 (25.0)	7 (14.3)	0.08
		1	10 (58.8)	5 (25.0)	22 (44.9)	
		2	1 (5.9)	7 (35.0)	15 (30.6)	
		3	0 (0.0)	3 (15.0)	5 (10.2)	
		No data	2 (10.5)	0 (0.0)	1 (2.0)	
Smoking status (%)		Never smoked	3 (15.8)	2 (10.0)	0 (0.0)	0.016
		Former smoker	0 (0.0)	3 (15.0)	8 (16.0)	
		Current smoker	2 (10.5)	7 (35.0)	18 (36.0)	
		No data	14 (73.7)	8 (40.0)	24 (48.0)	
Symptoms (%)		No symptoms	3 (15.8)	1 (5.0)	0 (0.0)	0.023
		Hoarseness	0 (0.0)	1 (5.0)	3 (6.0)	
		Dyspnea	10 (52.6)	9 (45.0)	14 (28.0)	
		Cough	1 (5.3)	1 (5.0)	5 (10.0)	
		Hemoptysis	2 (10.5)	6 (30.0)	24 (48.0)	
		Other	0 (0.0)	1 (5.0)	3 (6.0)	
		No data	3 (15.8)	1 (5.0)	1 (2.0)	
Narrowing of the tracheal lumen (%)		≤49%	3 (15.8)	7 (35.0)	11 (22.0)	0.302
		≥50%	10 (52.6)	7 (35.0)	30 (60.0)	
		No data	6 (31.6)	6 (30)	9 (18.0)	
TNM	T (%)	1	9 (52.9)	1 (7.7)	15 (35.7)	0.035
		2	8 (47.1)	12 (92.3)	27 (64.3)	
		No data	2 (10.5)	7 (35.0)	8 (16.0)	
	N (%)	0	15 (88.2)	6 (46.2)	20 (48.8)	0.014
		1	2 (11.8)	7 (53.8)	21 (51.2)	
		No data	2 (10.5)	7 (35.0)	9 (18.0)	
	M (%)	0	17 (89.5)	14 (70.0)	46 (92.0)	0.047
		1	2 (10.5)	6 (30.0)	4 (8.0)	
		No data	0 (0.0)	0 (0.0)	0 (0.0)	

^1^ To examine the significance of the association, Fisher’s exact test was used for categorical data and the Mann–Whitney U test was used for continuous data.

**Table 4 cancers-14-01665-t004:** TNM clinical characteristics of 89 patients diagnosed with primary tracheal cancer by treatment intention.

TNM		Palliative Treatment	Radical Treatment	*p* ^1^
Number of patients		44	45	
T (%)	1	9 (28.1)	16 (40.0)	0.422
	2	23 (71.9)	24 (60.0)	
	No data	12 (27.3)	5 (11.1)	
N (%)	0	15 (45.5)	26 (68.4)	0.087
	1	18 (54.5)	12 (31.6)	
	No data	11 (25.0)	7 (15.6)	
M (%)	0	33 (75.0)	44 (97.8)	0.005
	1	11 (25.0)	1 (2.2)	
	No data	0 (0.0)	0 (0.0)	

^1^ To examine the significance of the association, Fisher’s exact test was used for categorical data and the Mann–Whitney U test was used for continuous data.

## Data Availability

Data may be available upon reasonable request and with permission of the National Research Institute of Oncology, Warsaw, Poland.

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
