# Peer review of "Do We Need TNM for Tracheal Cancers? Analysis of a Large Retrospective Series of Tracheal Tumors"

_cancers, 2022, doi:10.3390/cancers14071665_

Round 1

Reviewer 1 Report

The authors did a good job I think and answered all my comments in the manuscript. Their paper is also edited based on my review, so overall I am satisfied with this manuscript. I think we should accept their manuscript. 

Reviewer 2 Report

Thank you for your efforts. The authors appropriately addressed all of my remarks. I find the paper suitable for publication.

This manuscript is a resubmission of an earlier submission. The following is a list of the peer review reports and author responses from that submission.

Round 1

Reviewer 1 Report

The authors presented a novel approach to TNM classification of tracheal cancers. THe manuscript is well-written and easy to read, there are only few typos. The topic has realistic importance in the treatment of tracheal cancer.  The main limitations were the retrospective nature and the small patient number. Although, there are some limitations in the study, the patients' OS results can support their suggestion regarding TNM evaluation.

The T category was only determined with the anatomical position of the primary tumor, it would be advisable to add tumor size. What was the exact staging method? Did the authors take imaging ( e.g. PET-CT) or endoscopic procedures ( bronchoscopy or ultrasound guided biopsy)?

Why was this classification the same in all type of histology? Both clinical and pathological differences can influence the TNM in each tumor type. It should be better to do the classification separately for each histological type. 

Regarding the Figure 1 and 2, the OS can be different in each histological type. Did the authors check if there any differences between the OS of ACC vs. SCC T1/T2, N0/N1/, M0/M1 patients? 

In Table 2 it is hard to follow which groups had significant differences. To what exactly the p values refer to? Because there are 3 groups, and more options of WHO performance status or smoking status, but there is only one p value. It is not clear what the authors compared. Similarly, when the authors compared palliative vs. radical treatment in table 3, there is one p value for 3 T, N and M subgroups ( 1, 2, no data). 

Which regional lymph nodes were effected? Is there a difference in the OS of patients based on the anatomical position of regional lymph node metastasis (pre-tracheal, paratracheal etc) or in the numbers of affected lymph nodes?

The authors TNM classification was unable to calculate with size of the primary tumor that would be really important. The N and M category should be also improved. Did the authors compared those patients who had only 1 distant metastasis vs. those who had multiplex metastases? It may also influence the further therapeutical options. 

Minor comment: Fig 1, 2,3 'nodal involvment' instead of involvement.

Reviewer 2 Report

Recently I was invited to review an interesting paper entitled “Do we need TNM for tracheal cancers? Our own results and literature review”. The authors present one of the largest series in the literature. It is important to discuss the perspectives of staging and treatment even in rare neoplasms. The paper is well written and I do not have significant remarks concerning the scientific style or the quality of English. The graphs and figures have some flaws which may be easily corrected. Nevertheless, I have some major remarks that should be addressed prior to the publication of the paper.

Major remarks.

Lines 2-3. The title is catching and I like it. However, please consider rephrasing the second sentence. Results are usually both “our” and “own”. A literature review is a part of every paper. On the other hand, you present one of the largest retrospective series in the literature. This is an advantage of the paper. Why not use that in the title.

Lines 77-79. Please provide a clear inclusion/exclusion flowchart presenting how many patients were excluded due to precisely described reasons.

Table 1. In the T2 descriptor. Please specify the involvement of which structures would be enough to be included in T2. Esophagus, thyroid gland – easy to understand. How about the muscles, fat, or recurrent nerve? This should be described in a more detailed way.

Lines 106-107. How many patients were lost to follow-up? What were the methods of follow-up?

Table 2. The data about the smoking status are characterized by a significant percentage of missing data. In case of such a poor quality of this data, please consider removing it from the table.

Table 2. There is a significant number of patients with “other” types of tracheal malignancy. This is a very interesting group. Please consider specifying this group in a separate table disclosing all of the rare histologies.

Lines 130-135. What were the methods of an initial assessment? Probably all of the patients had chest and neck CT. How many patients had PET-CT and MRI? If this information is not available, please state that clearly, then again in study limitations.

Lines 142-149. Patients were treated in a single center. Were all of them consulted by a multidisciplinary team including a thoracic surgeon, ENT specialist, and radiation therapist? If this information is not available, please state that clearly, then again in study limitations.

Line 153. In the Table 2 authors use 3 decimal places to precise the p-value. It is sufficient. However, in the text authors use 4. Please unify that throughout the text (preferably 3 is enough).

Figures 1, 2, 3. Please add numbers of patients at risk below every analysis.

Lines 196-262. In the Discussion, (especially T descriptor) I find not enough critical argument of the results obtained by the authors. Why do the authors think that their proposal of the TNM system is better/worse than the previous ones?

Lines 196-326. Discussion. Please specify what are the limitations of the paper. Moreover, please specify the authors' expertise about the future expansion of the system. What could be the methods for that? Future collaboration on the basic projects including national or international databases seems to be interesting. The TNM of lung cancer is created mainly on the basis of the data accumulated in surgical databases. Do the authors think that the role of thoracic surgeons is important in this part of a project? Would it be valuable to collaborate with scientific societies like ESTS, STS, EACTS in this kind of project?

Lines 327-337. The first seven sentences are no real conclusions. That should be moved to the Discussion. The only conclusion is the last sentence. Please expand this sentence and present condensed results in 2-3 additional sentences. The authors' results are so clear and well documented. Do not hesitate to do that.

Last remark. You open the paper with a question placed in the title. Maybe it would be worth to end the paper with an answer?

I would like to congratulate the authors on their efforts. It was a pleasure to review this valuable work.
